# Experiences of Cultural Differences, Discrimination, and Healthcare Access of Displaced Syrians (DS) in Lebanon: A Qualitative Study

**DOI:** 10.3390/healthcare11142013

**Published:** 2023-07-12

**Authors:** Riwa Khalifeh, William D’Hoore, Christiane Saliba, Pascale Salameh, Marie Dauvrin

**Affiliations:** 1Institute of Health and Society, UCLouvain, 1200 Brussels, Belgium; william.dhoore@uclouvain.be (W.D.); marie.dauvrin@uclouvain.be (M.D.); 2Faculty of Public Health—Section 2 (CERIPH), Lebanese University, Fanar 90656, Lebanon; christiane.saliba@ul.edu.lb; 3School of Medicine, Lebanese American University, Byblos 1401, Lebanon; pascalesalameh1@hotmail.com; 4Institut National de Santé Publique d’Épidémiologie Clinique et de Toxicologie-Liban (INSPECT-LB), Beirut 1103, Lebanon; 5Department of Primary Care and Population Health, University of Nicosia Medical School, Nicosia 2417, Cyprus; 6Faculty of Pharmacy, Lebanese University, Hadath 1533, Lebanon

**Keywords:** Syrians refugees, health equity, discrimination, access to health services, Lebanon, qualitative study

## Abstract

The study aims to examine cultural differences and discrimination as difficulties encountered by DS when using the Lebanese healthcare system, and to evaluate the equity of DS access to health services in Lebanon. This is a qualitative study using in-depth semi-structured interviews with DS and Lebanese healthcare professionals. The participants were selected by visiting two hospitals, one public Primary Healthcare Center, and three PHCs managed by Non-Governmental Organizations. The recruitment of participants was based on reasoned and targeted sampling. Thematic analysis was performed to identify common themes in participants’ experiences of DS in accessing Lebanese healthcare. Twenty interviews took place with directors of health facilities (*n* = 5), health professionals (*n* = 9), and DS (*n* = 6) in six different Lebanese healthcare institutions. The results showed barriers of access to care related to transportation and financial issues. Healthcare services provided to the DS appear to be of poor quality due to inequitable access to the health system, attributable to the discriminatory behavior of healthcare providers. Among the several factors contributing to the presence of discrimination in the Lebanese healthcare system, the persisting fragility of the healthcare system—facing a humanitarian crisis—emerged as the major driver of such unequal treatment. The number of DS in Lebanon is roughly equal to a quarter of its citizens; there is an urging need to restore the Lebanese health system to ensure the equitable provision of health services for DS and appropriate working conditions for health professionals.

## 1. Introduction

In recent years, migration increased significantly to an unprecedented level, with a documented 3.5% of the world population now living outside their native land [1], creating ethnically and racially diverse communities in various parts around the world [2].

In 2011, the Syrian civil war led to a massive influx of Syrian refugees to Lebanon due to its geographical proximity [3,4]. Lebanese government data indicate that the country hosts 1.5 million DS, equivalent to a quarter of the Lebanese population [5]. However, in view of the Syrian hegemony over Lebanon from 1975 to 2005, the history of troubled relations with Syria has marked Lebanon and its inhabitants (of approximately 4 million) [6]. Moreover, due to historical and legal reasons closely connected to the age-old Palestinian presence in Lebanon following the Israeli-Palestinian conflict in 1948 and the consequences of their displacement and settling down on Lebanese grounds, the Lebanese government—which has never ratified the 1951 Geneva Convention—has always refused to grant refugee status to Syrian populations fleeing the Damascus regime [6]. Hence, Lebanon has officially used the word “displaced” to designate populations seeking protection and has always refrained from using the term “refugee” [6].

Studies show that migrants and refugees worldwide need significant healthcare services and suffer from disparities and health inequities in their access to care [7,8,9]. Healthcare biases and cultural differences in language, beliefs, or values are the main causes of healthcare disparities [10]. Bias in healthcare is commonly defined as the tendency to favor one group over another [11]. Therefore, discrimination is characterized as bias when it results in unequal treatment based on a person’s physical traits or membership in a certain social group [12]. Healthcare bias can be either positive or negative and can be either unconscious (implicit or unintentional) or conscious (explicit or purposeful) [11]. However, health equity remains a right, regardless of the status of the care recipient, and an essential element in the strategy of the health sector [13].

In the Lebanese context, citizens do not have sufficient access to healthcare services given the poorly structured system. Both the provision of healthcare services and their finance remain mostly in the hands of the private sector and the responsibility of the individual. The three primary mechanisms of finance for the Lebanese healthcare system are public or fiscalized, private, and parapublic or socialized. The government’s direct action is still insufficient, especially for those who are most in need [14]. The impact of the Syrian conflict, the Lebanese economic crisis (since October 2019), and the Beirut blast (August 2020) have taken their toll on the Lebanese healthcare system. Such strain has led to an increased demand for healthcare services, the accumulation of unpaid debts incurred by the Ministry of Public Health to hospitals under contract, a shortage of health professionals such as medical specialists and nurses, an outbreak of communicable diseases among the ‘displaced’, and the emergence of new diseases [14].

At the beginning of the Syrian crisis, the Lebanese health system showed resilience and responded to DS’s needs for healthcare. The equitable access to health services granted to Syrians is due to the Lebanese Ministry of Public Health’s (MoPH) collaboration with international organizations, the private sector, and civil society organizations [15]. Despite all efforts, barriers to the integration of the DS in the Lebanese health system persist due to the complexity of the health system in Lebanon added to the high cost of care, health inequity, socio-cultural disparities, and low interest from the provider’s perspective [16,17]. There are several studies describing the Lebanese and Syrian beneficiaries’ access to health services [14,17,18,19], but there are no studies capturing health disparities from the perspectives of DS in Lebanon. 

Moreover, DS, who make up 25% of the overall population, are blamed for escalating political and economic problems and challenges [20]. 

The aim of this study is to explore difficulties encountered by DS while accessing the Lebanese healthcare services, the relationship of Lebanese health professionals with DS and their attitudes while providing them with health services, and the effects of DS’s equitable access to health in a collapsing health system in Lebanon.

## 2. Materials and Methods

### 2.1. Study Design and Sample

We used an exploratory, descriptive qualitative design. The study was conducted in health facilities (PHCs, hospitals) located in six Lebanese governorates for a period of three months, from June 2020 to August 2020. Participants were managers, health professionals, and DS. We used reasoned and targeted sampling to recruit participants, based on the presence of healthcare professionals and DS in the facility at the time and day of the scheduled interview. 

Managers and health professionals were selected in hospitals and PHCs that receive DS. We visited two hospitals (one public and one private), one PHC, and three NGOs delivering care through PHCs. DS were selected and interviewed during the onsite visits. We contacted the healthcare facilities by phone prior to the visit and arranged meetings with the directors. The interviews took place in a space set aside for the purpose in the healthcare facility. There was no incentive for participation.

Inclusion criteria were the following:

Category 1: Administrators of health facilities were defined as directors and managers at managerial and administrative levels in a public or private health facility in the different Lebanese governorates which provide health services to displaced Syrians registered or not with the United Nations High Commissioner for Refugees (UNHCR).

Category 2: Health professionals were defined as certified Lebanese physicians and nurses with over two years’ experience in providing healthcare to DS in the health facilities described above and in providing either ambulatory care (PHCs) or hospital care.

Category 3: DS were defined as Syrian citizens that fled the civil war in Syria and settled in Lebanon in refugee camps or in private housing and who are entitled to benefit from health services provided by healthcare facilities described above whether or not they are registered or with UNHCR. 

The final study sample consisted of 20 participants (see Table 1). Five participants were medical doctors (*n* = 2) or managers or directors of healthcare facilities with a master’s degree in hospital management (*n* = 3). Nine participants were healthcare professionals working in selected healthcare facilities: five nurses and four medical doctors. Six DS—four women and two men—participated in the study during their visit to the selected healthcare facilities.

### 2.2. Interview Guides

Three semi-directive interview guides were formulated in French based on the objectives presented in Table 2, and then translated into Arabic using backward and forward translations as a method. 

The interviews were conducted in Arabic. The translation in both French and English languages was ensured by professional trilingual translators and then confirmed by the research team. This triangulation guaranteed the credibility of the data translation and the fidelity to the cultural sensitivity of the questions. The duration of the interview ranged between 20 and 40 min (average = 30 min). 

### 2.3. Ethical Approval

Participation in the semi-structured interviews was voluntary. The participants were informed about the purpose of the study and gave verbal consent for recording the interview. The institutional review board of Al Rahma Hospital (Ref: 077-2021) was informed and consented to conduct the study.

### 2.4. Data Analysis and Trustworthiness

A general inductive approach was used for the thematic analysis. This approach “provides a set of easy-to-use and systematic procedures for analyzing qualitative data that can produce reliable and valid results” [16]. An initial working coding scheme was generated from a consecutive review of the first set of transcripts by RK. Codes were developed through an initial open coding process, whereby codes were derived from the raw data. Data were also categorized in light of the research aims and questions that guided the development of interview guides. This working coding scheme was then applied to the same transcript by MD and WD to complete the coding scheme and ensure the validity of the coding scheme. The resulting coding scheme was then discussed and validated by all authors before being applied to the whole set of transcripts. The working coding scheme was revised until no new theme was identified. Each category was then reviewed for cohesion and compared to coding conducted by the other authors for consistency. The results were presented in the form of a narrative synthesis and were illustrated by excerpts taken from the interviews with the participants. The themes and sub-themes are elaborated below in Section 3.

Several measures were taken to ensure the trustworthiness of this study [21]:1.The research question was carefully crafted by the first author and thoroughly reviewed by the research team to ensure that it was clear and precise.2.A diverse group of participants was selected for the study, including directors, healthcare professionals, and displaced Syrians, to ensure that a variety of viewpoints was reflected. We referred to the person triangulation and collected data from 3 types of people, with the aim of validating data through multiple perspectives.3.The interviews were conducted in a neutral and non-judgmental manner, with open-ended questions that encouraged participants to share their experiences and opinions.4.The interviews were conducted by a trained researcher who was skilled in qualitative research methods and who had experience working with diverse populations. A peer debriefing was used with knowledgeable peers and researchers on a qualitative basis.5.The data collected from the interviews was carefully analyzed and interpreted using established research methods, including coding and thematic analysis. The findings were then reviewed and validated by multiple researchers to ensure that they were accurate and reliable, reflecting the experiences and perspectives of the participants.6.Data saturation was reached once no new information was obtained and redundancy was achieved.

## 3. Results

### 3.1. Access and Entitlement of the DS to Lebanese Health System Services

#### 3.1.1. Conditions for Hospital Admission

According to the interviews conducted with the directors and healthcare professionals, for the admission of DS to hospitals, the administrative staff should obtain the approval of a third-party payer designated by UNHCR for health services coverage and reimbursement. The patient co-pays 25% of the hospitalization costs. Childbirth is covered, taking into account registration with UNHCR, and many exemptions apply, including chronic diseases, infectious diseases requiring long-term treatment, surgeries with implants, and many others. Sometimes, the staff of the hospital call security if the patient refuses to pay the requested amount related to the healthcare provided during his/her hospitalization.

“*UNHCR is represented by Next Care, an insurance company that covers hospitalization costs. But some cases are exempted such as chronic diseases, orthopedic prostheses, any surgery considered non-urgent (cold case) […] For infectious diseases requiring long-term treatment, the patient will not be admitted to the hospital if he cannot cover the expenses out-of-pocket since these cases are exempted as per the UNHCR agreement. For the amount that the patient must co-pay, we used a new procedure that requires the patient to cover these expenses upon admission to avoid any conflict upon discharge […] Honestly, sometimes, we call on the Security to make the patient pay the requested amount*”(Jacques*—fictious name, director of a public hospital in Mount Lebanon).

The MoPH does not reimburse outpatient services. It offers an alternative by subsidizing a comprehensive package of PHC services through an extensive network of PHCs. DS registered or not with the UNHCR can benefit from the various treatments offered in these centers. Directors and healthcare professionals reported that the DS are unorganized, and they do not respect the consultation times and call doctors at any time of the day or night. These aspects may reflect, based on their statement, the way they behave in their country.

“*It took me a huge effort to explain and make them understand the organization of care in our center and the importance of respecting consultation times*”(Farida, nurse at a PHC in Mount Lebanon).

“*The DS are not organized […] besides, you can’t imagine, they believe that they can call the doctor at any time...it’s crazy, even for trivial questions, they call midnight. Frankly, I don’t know if they do the same thing in Syria […] they don’t understand that doctors are not employees who live in hospitals 24 h a day*”(Jacques, director of a public hospital in Mount Lebanon).

As explained by hospital directors, due to delays in third-party payers’ reimbursement, they decided not to admit DS until the pending invoices are paid.

“*Third-party payers are always late to finance care for DS […]. So, sometimes I make a decision not to have DS admitted unless they put a deposit before their admission*”(Jacques, director of a public hospital in Mount Lebanon).

#### 3.1.2. Access of DS to Health Services

As reported by the DS, challenges to access health facilities and receive the needed health services are related to transportation and financial difficulties to co-pay the amount not covered by the third party. 

“*Sometimes I don’t have the money to go to the center; I stay at home*”(Ahmad, Syrian, 25 years).

“*If they ask me for check-ups and x-rays, I don’t do them…I don’t have the money and I don’t come back to the center*”(Fatima, Syrian, 27 years).

According to DS, it is difficult to visit a doctor since there is no means of transport and the PHCs are far from their place of residence.

The interviews were conducted during the Coronavirus Disease 2019 (COVID-19) pandemic; participants reported avoiding consulting with a physician, as they feared being asked to self-isolate or to be admitted to the hospital, without having possibilities of contacts with their relatives. The poor living conditions and the lack of access to proper hygiene infrastructure appeared to trigger this fear. 

“*The other time, a doctor told me I have to stay at home because I have COVID symptoms and it is contagious, but in the tent where I live we are 20 people […] I am not returning to this doctor and if I know I will be consulted by him again, I will flee*”(Mohamad, Syrian, 32 years).

A director of a PHC mentioned an additional challenge with DS related to health literacy and difficulties using healthcare services. To compensate for this barrier, information material is put at disposal of the DS. 

“*Brochures and information in Arabic are offered in the PHC centers to allow them a better understanding of the health procedures carried out in these centers*”(Joseph, director of a PHC).

A director of a public hospital also stated that DS do not want to understand these procedures and refuse to be involved in the care and take part in decisions about the care received. 

“*…It’s not that they don’t understand. The problem is that they don’t want to understand!*”(Milad, director of a public hospital in Mount Lebanon).

To better facilitate access to healthcare services and to avoid legal barriers, one of the health professionals explained:

“*In our center (PHC-Hermel), we welcome any DS without asking for their identity or any legal document; you know that there are many Syrians who do not have their legal papers and who are sometimes afraid to go to the centers because they may be arrested by the Lebanese State*”(Ali, nurse working at a PHC).

### 3.2. Socio-Cultural Interplays and Interactions between DS and Lebanese Healthcare Professionals

#### 3.2.1. Socio-Cultural Differences

In their interviews, managers and health professionals pointed out that DS have different ideas of health, life, being a parent, treatments, and death than they do. In the Lebanese interviews, the beliefs and health attitudes of DS are described as “unusual” and “strange”. 

These cultural differences appear to be intertwined with social and economic considerations, making it even more surprising or shocking for Lebanese professionals who witness it. According to the interviews conducted with directors and healthcare professionals, issues relating to gender, religion, and the relationship between men and women create tensions between Lebanese healthcare professionals and DS. One example provided by a director of a hospital shows that DS may refuse to take their baby from the hospital because of the gender or even force their wife to have an abortion for the same reason. He also added that a Syrian father refused to pay the requested amount for healthcare provided to his little son, justifying his refusal by the fact that he can have another one.

“*One of the DS wanted to leave his newborn baby in the hospital because he does not want to pay the amount requested […]. Another left his wife and baby in the hospital because the woman gave birth to a girl and not a boy; it’s weird though. Also, a husband has asked his wife to have an abortion and he doesn’t care because he can have another one (he doesn’t want to pay…). I can also tell you that once we informed a father that his little one needed to be admitted to intensive care and there was an additional charge to be paid; he refused and replied, “it doesn’t matter if she dies; I don’t want to pay and I can have another” […] It’s still a bizarre attitude and mentality*”(Jacques, director of a public hospital in Mount Lebanon).

DS also experience these socio-cultural differences in their relationship with Lebanese physicians and appear more satisfied with healthcare provided by Syrian professionals than Lebanese ones. But the financial component also appears to play a role in their perceived satisfaction. 

“*The Syrian doctor understands us; he gives us an injection in his clinic, and everything goes well […] here, the doctors ask for tests, and we don’t have the money to do them*”(Ali, Syrian, 28 years).

It appears from the interviews that health professionals consider that the DS do not follow the recommendations in connection with the treatment or the use of the care services. Consequently, health professionals view Syrian patients as nonchalant and unorganized. 

Because of these differences and a lack of knowledge of the Lebanese healthcare system, Syrians may exhibit attitudes that are negatively perceived by healthcare professionals, such as showing up without respecting the appointment or neglecting the preventive measures or the advice recommended by Lebanese professionals. 

#### 3.2.2. Language Barriers

As experienced by healthcare professionals, language constitutes a barrier when DS speak a Syrian dialect or have a particular accent. The dialect negatively affects the communication between the healthcare professionals and the patient, and subsequently influences the quality of care and health outcomes. 

“*It is very difficult to understand what they are saying, I am not familiar with their dialect… you know when we receive a Syrian who comes from the camps of Akkar, he speaks the “badawiyé” accent; frankly I do not understand anything; I make an effort to understand the signs and symptoms to know what to do. In Tripoli (northern Lebanon), it is easier since there is another community, and there are similarities between the two Lebanese and Syrian populations who share the same customs and values in general*”(Fatima, nurse working at a PHC).

For the DS, as reported in the interviews, the language barrier is due to the use of scientific language. Consequently, patients reported not understanding the information but did not mention it to the professionals.

“*Sometimes the doctor uses foreign language words or uses Lebanese words that I don’t understand, but I say nothing to him, and I pretend that I have understood, and I leave…*”(Yusra, Syrian visiting a PHC in Akkar).

### 3.3. Health Inequity and Discrimination

Across the interviews, it appears that numerous interactions between DS and Lebanese professionals are reported in negative terms. In the professionals’ discussions as well as in the DS interviews, stereotypes, prejudices, blaming, and testimonies of inappropriate behaviors could be found. Whether intentional or unintentional, it was observed that these prejudices led to inappropriate care, poor quality of care, delays in seeking care, segregation in the services, aggressive reactions, as well as refusal of care. 

“*We sometimes encounter a problem of discrimination; especially with doctors who have a difficult character to manage… well you know they (doctors) consider that they occupy a particular position and that nobody can discuss it with them… We have received a lot of complaints from displaced Syrians about the sometimes inappropriate behavior of doctors. For example, one of the Syrian patients comes to tell me that a doctor told him to go to Syria to benefit from the care and that he gave up his place for a Lebanese.… Oooh what do we see cases, even doctors who refuse to see Syrian patients (only because he is Syrian), do you realize that?*”(Fatima, nurse working at a PHC).

Following the interviews, DS reported being rejected and/or discriminated against while receiving healthcare services. The stereotyped behavior of health professionals induced ambivalent feelings among DS, thus leading to them discontinuing the care.

“*The Lebanese always tell us that these centers are theirs and that we shouldn’t be here. We must return to our country*”(Jafar, Syrian visiting a PHC).

Sometimes, DS avoided coming back to the center because of this perceived feeling of rejection. They said that health professionals speak to them in a threatening and aggressive way and sometimes even leave them waiting without a valid reason. 

“*I come to this center because they behave better with us. The other time I was in a hospital, the doctor spoke to me in a disrespectful way and the nurse left me in the hallway for an hour without her speaking to me, yet she let other patients pass (Lebanese) smiling to them*”(Yusra, Syria, 32 years).

For their part, the displaced Syrians, following the interviews, expressed themselves regarding the rejection they felt from Lebanese health professionals, which constitutes a barrier in the provision of care. The stereotyped behavior of health professionals induces ambivalent feelings among displaced Syrians and thus leads to a halt in the continuity of care: *“The Lebanese always tell us that these centers are theirs and what are we doing here? We! We must return to our country”.*

Discriminative practices could be found in the professionals’ attitudes but also among Lebanese patients attending the same services, leading to a physical separation of patients inside the services. 

“*We had a lot of resistance from the staff at the hospital to work with the Syrians. The nurses complained about the smell and the hygiene of the Syrian patients (remember that they live in tents with miserable conditions) which pushed them to do things in a hurry and sometimes walk past symptoms without considering them. Following the various complaints received from patients (Lebanese), we were forced to put Syrian patients in separate rooms*”(Nesrine, head nurse working at a hospital).

In the interviews with the healthcare professionals, these discriminant attitudes seem to find their roots in the sociopolitical context surrounding the relations between Lebanon and Syrian, notably the Syrian hegemony over Lebanon until 2005. Besides, as reported by some professionals, Lebanon is still recovering from the Lebanese civil war and has to cope with scarcity of resources, materialized, among others, in the collapsing Lebanese health system. The healthcare system struggles with a lack of human and financial resources. These factors seem to promote discrimination and prejudice in healthcare but could be seen by some professionals as the expected consequences of the shared war history. 

“*To be frank with you, yes, there is discrimination between doctors/nurses and DS…it’s very human and don’t forget that we are Lebanese who lived through a war with Syria and we still have this feeling that they are “Syrians”*”(Mohamad, doctor consulting in a PHC).

#### Factors Contributing to Health Inequity and Their Effects on DS in the Context of Lebanese Health System

Based on the different themes that emerged, the health inequity of DS in Lebanon is related to socio-cultural and financial interplays and the lack of understanding of the Lebanese health system. The history of war stigma has created everlasting tensions and seems to be a leading cause of discrimination and difficult interaction with the Lebanese healthcare professionals. The transportation to care facilities and the financial problems that limit the ability to co-pay the extra health costs not covered by third-party payers are barriers to access health services. Our study highlighted other factors contributing to health inequity, like the continuity of care, health literacy, and legal issues. Figure 1 shows that the drivers of discrimination are stereotyping, prejudices, stigma, and racism. 

## 4. Discussion

By examining the experiences of DS and Lebanese health professionals, this study reveals many barriers and difficulties that stand in the way of DS accessing healthcare. Its findings are consistent with those of other studies at the international and national levels. 

The results of this study show that transportation to care facilities as well as financial problems due to limited ability to co-pay extra health costs (not covered by third-party payers) constitute barriers for DS’s access to health services. This aligns with previous studies confirming that high costs of services and lack of confidence in the Lebanese system hinder access to health services for DS [17,19]. Thus, the underfunded health system combined with weak human and financial resources contributes to DS’s reduced access to personalized care. Furthermore, the COVID-19 pandemic hindered the access of DS to healthcare facilities because of fear of getting isolated. This was closely linked to the poor living conditions of DS and the difficulty they had with self-isolation, while living in a tent crowded with big family members. This lines up with another study revealing how DS neglected preventive practices of COVID-19 due to their harsh living conditions [22]. 

In addition, communication barriers and lack of or incomplete health literacy sometimes led to misdiagnosis and even worse outcomes. Although Lebanese and Syrian people speak a common language, they use different Arabic dialects, which may sometimes negatively affect their comprehension and result in poor health outcomes. According to several studies, there is a link between language barriers, health literacy, and bad health status among refugees [23,24]. Developing mutual trust, eliminating uncertainty, and tailoring messages to DS’s health literacy levels can promote a healthy intercultural climate [25] and foster the interaction with the Lebanese health system [26]. 

The DS’s socio-cultural perception of the Lebanese physicians is revealed by some financial considerations and unaffordable prescriptions. It is worth mentioning that the prescription of some medication in medical centers and hospitals in Lebanon abides by the requirements of MoPH, which restricts giving certain medication without carrying out a full medical test. This seems to influence the satisfaction of DS and their compliance to treatments. Previous studies pointed to the socio-cultural hurdles that prevent migrants and refugees from seeking emergency or medical care, thus resulting in poor healthcare [27,28,29]. Additionally, DS seemed more confident when interacting with Syrian medical professionals. This supports earlier research which demonstrates that when dealing with healthcare experts of their own nationality, migrants were more satisfied and compliant with their treatment [29].

More particularly, the findings of this study shed lights on the cultural issue of gender among DS. The findings described the type of interaction between Syrian men and women and highlighted the power and the dominance of men over women, which is perceived as natural and normal [30]. In this context, the literature suggests that healthcare professionals need to acquire intercultural care skills to counter the cultural barriers in healthcare [31]. For instance, the results showed a cultural particularity of the patriarchal right to determine the fate of children’s life. In fact, one father clearly expressed his right to potentially determine the end of his son’s life and justified it by his ability to have another child. 

Furthermore, the data of this study highlight the health inequity of DS in Lebanon. Discrimination can be an underlying mechanism that translates neglect and lack of concern into the practices of Lebanese health professionals, with various impacts on health outcomes for the DS population. The history of war stigma has created everlasting tensions between the two peoples, seems to be a leading cause of discrimination against the DS, and has often led to difficult interactions with Lebanese healthcare professionals. 

This is consistent with the findings of other research demonstrating that the complexity of the health system in Lebanon, its high costs, recurrent biases among healthcare professionals, and a general lack of interest and concern on the part of physicians are the main obstacles that prevent easy access for DS to healthcare in Lebanon [17,18]. 

Moreover, the results of this study reveal that the presence of DS seems to generate responses and feelings of ambivalence and hostility among Lebanese health professionals who have contributed to difficulties in DS accessing healthcare. The literature reveals that health inequity, whether intentional or unintentional, may influence the patient’s perception, behavior, and interaction, and may alter his judgment [32]. It affects vulnerable people such as ethnic minorities, sexual minorities, immigrants, the poor, the mentally ill, the disabled, and anyone who is vulnerable to some context [33,34]. However, non-discriminatory access to health services is one of the fundamental principles of health law. Similarly, equal access to healthcare is not limited to physical restrictions but to the provision of treatment. Research from the United States has documented rates of discrimination in healthcare settings based on race, ethnicity, immigration status, language, and insurance status [35,36,37]. In this context, several studies showed a close link between experiences of discrimination in healthcare settings and delayed or abandoned care in both the United States and Europe [37,38]. Discrimination in healthcare settings can lead to lack of trust and satisfaction with the healthcare system [28]. In the Lebanese context, our study showed that DS feel marginalized, discriminated against, rejected, and judged. Parallel investigations revealed that prejudice and discrimination towards DS in Lebanon has a negative influence on their ability to access resources, particularly the healthcare system [39,40].

### 4.1. Strengths and Limitations of the Study

To our knowledge, this is the first study that has targeted health inequity in the provision of health services to DS that includes a perspective of the DS themselves. The results are compatible with the set criteria of qualitative research. The sample size is not intended to be representative but rather aimed at covering the diversity of the situations encountered in the Lebanese healthcare system. One limitation may be that the objectives of the interview guides were targeted to the experiences and difficulties of the DS and the Lebanese healthcare professionals who are offering the care to them. However, the results of the interviews highlighted the theme of discrimination and reported the health inequity of DS when they are seeking health services in Lebanese health facilities. Despite these limitations, the understanding of stigma and/or discrimination in healthcare settings has been enriched in a resource-rich setting. This is the first qualitative study of the discrimination perceived by DS in the Lebanese healthcare settings. Furthermore, interviews were conducted by the first author, who had an in-depth understanding of the Lebanese healthcare system and the particular context where the study was carried out. It is worthy to mention that the first author tried and remained neutral, set aside her own views and reactions, and listened from the perspective of a researcher. In addition, we attempted to enhance transferability by thoroughly describing the research context and methods, and relating our results to existing evidence so that readers may better determine the relevance of these findings to other settings.

### 4.2. Future Research Directions

Our findings offer baseline data to give a general idea about the topic and help formulate hypotheses, anticipate, and better target the approach of the following interviews to answer new research questions. 

## 5. Conclusions and Recommendations

The number of DS in Lebanon is roughly equal to a quarter of its citizens. Lack of financial resources has resulted in discrimination in priorities for equity of care. Global logistic, technical, and financial initiatives can positively contribute to a radical and efficient change in the distribution of care resources. These initiatives include the use of free telemedicine and mobile clinics to provide healthcare services to DS. This can be particularly useful in rural regions and remote areas where access to healthcare facilities is limited. 

On the other hand, technical initiatives could include the use of health information systems that can provide real-time data on DS patients’ health status, enabling health providers to make more informed decisions about patient care. 

Moreover, there are financial challenges that need to be addressed to improve the distribution of care resources in the health system in Lebanon. The Lebanese health system infrastructure has collapsed, and the country is struggling with political, financial, and health crises. Therefore, we advocate for raising awareness of the health inequity of the DS to help Lebanon benefit from global funding and support to replenish the lack of human and financial resources. 

In addition, the lack of health literacy among DS in Lebanon is a serious problem that negatively affects their health outcomes, according to the results of this study. Many Syrians require proper knowledge and comprehension of fundamental health concepts. To address this issue, health literacy programs can be implemented to educate DS on topics such as disease prevention, hygiene practices, and accessing healthcare services. These initiatives can be carried out by community health workers and through medical facilities and training sessions. Furthermore, Arabic-language health literacy resources can be created and provided to DS via a variety of channels, including social media and community centers. By improving health literacy among DS in Lebanon, we can empower them to take control of their health and well-being and ultimately reduce health disparities among this vulnerable population.

To overcome cultural gaps between DS and Lebanese healthcare professionals and lessen healthcare inequities, it is essential to encourage cultural competency and sensitivity among healthcare professionals. This can be conducted by including courses on cultural competency in universities’ health programs for students majoring in medicine and paramedicine. Additionally, healthcare personnel providing treatment to DS should attend a training course on the cultural values, customs, and beliefs of DS.

Furthermore, universal health coverage could be a measure to address the financial insecurities the DS face as they most often cannot afford their medical expenses. This would secure their access to health services.

By implementing these initiatives, we can help improve the health system in Lebanon and ensure that all populations have access to the care they need.

## Figures and Tables

**Figure 1 healthcare-11-02013-f001:**
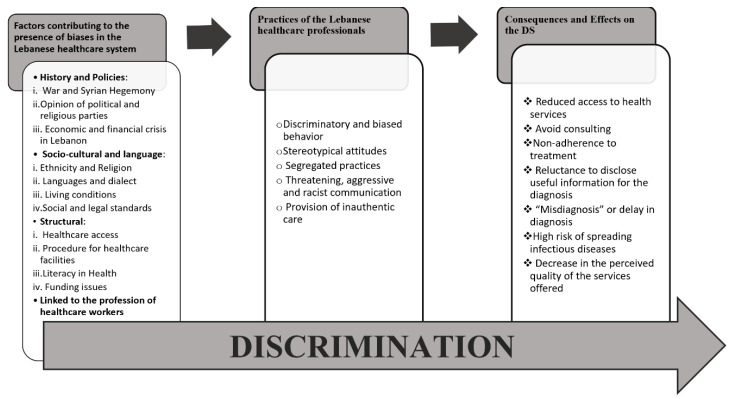
Factors contributing to health inequity and their effects on DS in the context of Lebanese health system.

**Table 1 healthcare-11-02013-t001:** Characteristics of the participants.

Participants	Code	Number of Participants (*n* = 20)	Participants’ Sex		Mean Age of Participants	Academic Qualification
			M	F		
**Category 1**: Administrators of Health Facilities	E_D_PX	5	3	2	47	Healthcare Management (*n* = 3), Doctors (*n* = 2)
**Category 2**: Health Professionals	E_PS_PX	9	6	3	38.5	Nurse (*n* = 5), Medical Doctor-General Practitioners (*n* = 4)
**Category 3**: Displaced Syrians	E_DS_PX	6	2	4	40.5	N/A

*n*: number; E: entretien/interview; D: director; PS: professionnel de santé/health professional; DS: displaced Syrian; P: participant; X: number of participants; N/A: not applicable.

**Table 2 healthcare-11-02013-t002:** Objectives of the interview guides.

Participants	Objectives of the Interview Guide
Administrators of health facilities	- To understand the conditions, required documents, reimbursement of cover costs, and other steps involved in the admission of DS to PHC and hospitals. - To comprehend the channels of communication and coordination between the different stakeholders: hospitals/PHC, MoPH, NGO, other actors, etc. - To know the main health needs of DS. - To describe the various problems and difficulties encountered by medical and paramedical personnel providing care to DS, including the relation and interaction between health professionals and DS.
Health Professionals	- To recognize the individual cultural factors of the DS, such as religion, dialects, conceptions of physical and mental health, life, and death, as well as beliefs about illness, health, and treatment… and try to understand the interaction and the effects of those factors in the delivery of healthcare services. - To describe the relation between the doctors/nurses and the DS and to comprehend how it affects the level of care provided to the patients.
Displaced Syrians	- To explore the difficulties encountered by DS while accessing the Lebanese healthcare system. - To understand whether DS receive appropriate care and services corresponding to their specific situations. - To describe the relation between the healthcare professionals and DS during the delivery of care.

## Data Availability

The datasets used and/or analyzed during the current study are available from the corresponding author on reasonable request.

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
