# Peer review of "Experiences of Cultural Differences, Discrimination, and Healthcare Access of Displaced Syrians (DS) in Lebanon: A Qualitative Study"

_healthcare, 2023, doi:10.3390/healthcare11142013_

Round 1

Reviewer 1 Report (Previous Reviewer 2)

Thank you very much for counting on me to review the work again. 

I think it has improved considerably. In this sense it has achieved sufficient quality to be published. Both the introduction, methodology and results have been greatly expanded, allowing for a better understanding and contextualisation of the work produced by the authors. Moreover, as I pointed out in my first review, the work uses an interesting qualitative perspective incorporating the point of view of the principal actors. 

I have only one issue of form: the format of table 1 needs to be revised, as although it is a simple and interesting table, it is difficult to read. 

Again, thank you very much.

Author Response

Dear Reviewers,

We are writing to express our sincere gratitude for your time and effort in reviewing our paper. Your constructive comments and recommendations have been invaluable in helping us improve the quality of our work.

We have carefully considered each of your suggestions and have made the necessary amendments to the paper. We believe that these changes have significantly strengthened the arguments presented in the paper and have helped to address some of the concerns that were raised.

We would like to take this opportunity to thank you for your thorough and thoughtful review. Your feedback has been instrumental in guiding us towards a more comprehensive and nuanced understanding of the subject matter. Your insightful comments have also helped us to identify areas where further research may be needed, and we are grateful for your guidance in this regard.

We would be happy to answer any further questions or concerns you may have about the revised paper. Thank you again for your time and consideration.

Sincerely,

The Authors,

Reviewer 2 Report (New Reviewer)

This is an interesting and challenging topic.  Generally speaking, the paper is well-written in terms of theory, data-driven and qualitative analysis (as the manuscript already improved).

The title of the manuscript impacts the reader; however, it would be necessary to have a short explanation of what means the discrimination experienced by the refugees in this particular context.

The introduction part is exhaustive and falls into contextual research.

The methods section (material and methods section) is generally written in a  concise manner.  A framework of themes and sub-themes might be helpful to have the overall picture of the social world of the displacement of refugees.  The theme ”health inequity and discrimination” could be extended in terms of explanations, theory, and triangulation data (in line with other related studies). It should be interesting to point out if adverse effects of discrimination may influence forms of acculturation or integration.

The conclusions are concisely presented and provide evidence for the issue addressed

Author Response

Reviewer 3 Report (New Reviewer)

Thank you very much for sharing this paper. I have several minor comments.

1. I think it is important to clearly define what discrimination means in this paper.

2. This paper talks a lot more than just discrimination - for example, this paper shows that culture and language issues. Perhaps the title and abstract cam reflect this complex nature of findings.

3. It may be useful to indicate that DS are more likely to be financially struggling compared to non-DS, making it more difficult to make health care accessible. There is always a structural condition. 

4. That being said, is it useful to discuss the role of universal health insurance as part of the recommendation section?

5. Overall, this is a wonderful study with many interesting findings. 

Author Response

This manuscript is a resubmission of an earlier submission. The following is a list of the peer review reports and author responses from that submission.

Round 1

Reviewer 1 Report

Dear Authors

I enjoyed your paper with the title: Experiences of discrimination and healthcare access displaced  Syrian refugees- a qualitative study.

The paper addresses an important topic and more research is needed on this access to care with immigrant populations.  The background is concise and well written and leads well into the rational why research is important.

I have recommendations to improve the paper.

1.       Check the number of participants as reported in the abstract. You said you conducted 20 interviews but the abstract you report  you have conducted 6+6+9 Interviews that equals 21 interview.

2.       You are using quantitative language to describe your design.  Using the term ‘cross sectional’ is an quantitative approach.  In qualitative design is is important to define which qualitative paradigm you have used for example phenomenology, ethnography, discourse analysis etc.

3.       Add the ethic approval number and the name of the specific ethics board that has approved your study. Some of your participants are from equity seeking populations and I am expecting more information how you ensure their privacy and confidentiality.  Did you give them an incentive to participate?  How long was the interviews and where did you conducted it?

4.       You stated that the interview guide was developed in French but that the interviews were conducted in Arabic.  How did you ensure the culture sensitivity of the questions?  Did you translate the interviews to English and how did you ensured the quality of the translation?  Have you used backward and forward translations as a method?

5.       How did you ensure that the study was trustworthy? 

6.       The headings used to describe the theme are not the same as the table that you added. 

7.       The discussion does not build on your findings. I am not sure how the inequities that you discussed are supported by your findings.  I suggest that you reconsider how your discussion are supported by your findings.  Consider to discuss your methods with an experienced qualitative researcher.

8.       I suggest that you add a comprehensive discussion on what you recommend for further research, practice and policy development. 

9.       I have also pick up on some spelling error.  Read the document again carefully for language and spelling edits.

The language is is off good quality.  the paper need another good read for spelling errors. 

Reviewer 2 Report

Thank you for the opportunity to review this interesting research titled “Experiences of discrimination and healthcare access of displaced Syrians (DS) in Lebanon: a qualitative study”.  The paper investigates the critical topic and an important issue for the health care system.

Findings based on various in-depth interviews among DS, directors and professionals provide various values insight. The work uses an interesting qualitative perspective incorporating the point of view of the principal actors. I have a few concerns that I would like the authors to address to improve the quality of the manuscript.

Introduction

Contextual information is needed. It is very important for readers and researchers to understand the context of this study and characteristics of its population. This can also help us gauge the findings.

The description of the Lebanon health system must be included.

The authors should be explained research gaps based on previous studies. It would add value of this study and how it differs from available ones.

The introduction must include the aims and the research questions that guided the proposed work.

Materials and Methods

It is not clear what the sample selection process was and how the participants were selected: were the centres telephoned, were they contacted by email, was there a facilitator? All this should be specified.

In the same vein, it would be necessary to incorporate other elements such as whether age, experience, time in Lebanon, gender or any other question that would help to understand the results. In my first commentary on the results there is a proposal in this regard.

It would be interesting to incorporate in more detail the dimensions and categories on which the three interview scripts are based. This would give us an idea of the content of the interviews. This would also give consistency and coherence to the work and the different sections, considering the proposal to incorporate the research objectives and questions.

The categories and codes resulting from the analysis are also unclear. Table A1 should be expanded to incorporate all the categories, subcategories and codes that have been derived from the analysis and which then seem to be reflected in the presentation of the results. This, again, would give consistency to the work.  That is, I would not only incorporate a summary of the themes, but I believe that table A1 should detail all the elements that are derived from the analysis described.

Results

The description of the sample would be included in the methodological part, detailing all the dimensions that have been part of the sample selection and that, as mentioned in the previous point, are not entirely clear. In this sense, it could be useful to use a table specifying the categories of participants, but also other relevant variables to understand and analyse the results (sex, age, time in each category, etc.). It would also include the name (pseudonym) of each participant.

Although the analysis carried out is interesting, I believe that more importance should be given to the discourses of the DS who are who face the barriers directly. That is, I believe that an important part of qualitative work is to give voice to the people affected by social inequalities, something that, in this work, although done, should be expanded.

For example, there is a central element that is a section of the results that is mentioned several times, and that is what has to do with barriers to access due to racism and discrimination. Despite its relevance, there is hardly any discourse from the affected DS. This needs to be addressed as it is one of the fundamental barriers faced by DS everywhere in the world.

In this regard, for example, in section 3.2.2 (Access of DS to health services), although it makes direct reference to DS, none of their speeches are incorporated.

I consider that the verbatims are very broad and that there is little analysis of the results. In other words, the authors leave the "responsibility" of analysing the discourses to the reader. I believe that the relevance of the selected discourses to address the categories that organise the results section should be further explored.

Discussion

The discussion should incorporate further reflection of the categories, incorporating more literature to understand the specifics of the results found in this paper. The discussion itself does not seem to be a discussion. I believe that the results should be analysed in the context of other work. I believe that, in this sense, a further review of the previous literature is necessary.  Perhaps there are no similar studies, but there is an extensive bibliography on the difficulties of inclusion, including health inclusion, of the migrant population and DS, including aspects related to the categories of work: language barriers, discrimination, socio-demographic aspects, cultural and economic differences, etc.

Going through each of the categories and subcategories, as well as collecting the objectives and research questions, I believe would help to deepen the discussion of the results.

At the bottom of page 9 the authors state that "In the Lebanese context, our study showed that DS feel marginalised, discriminated against, rejected, and judged". This cannot be derived from the analysis conducted. In fact, there is no evidence of any discourse by the DS themselves that reflects this.

Conclusions

They are scarce. I believe that a deeper exploration and discussion of the results will help to derive some more concrete issues. For example, what kind of global logistical, technical, and financial initiatives can positively contribute to a radical and efficient change in the distribution of care resources?

I hope that all the above will be useful to the authors to improve the proposed work.

Thank you very much for the opportunity.